behaviour/biomedical engineering/pattern recognition

behaviour, lameness, machine learning, sensor, signal processing, precision livestock farming

**Author for correspondence:**
Jasmeet Kaler
e-mail: jasmeet.kaler@nottingham.ac.uk

# Automated detection of lameness in sheep using machine learning approaches: novel insights into behavioural differences among lame and non-lame sheep

Jasmeet Kaler[1], Jurgen Mitsch[2],
Jorge A. Vázquez-Diosdado[1], Nicola Bollard[1],
Tania Dottorini[1] and Keith A. Ellis[3]

[1]School of Veterinary Medicine and Science, University of Nottingham, Sutton Bonington Campus, Leicestershire LE12 5RD, UK
[2]Advanced Data Analysis Centre, Digital Research Service, University of Nottingham, Nottingham NG8 1BB, UK
[3]Internet of Things Systems Research, Intel Labs, Leixlip W23 CX68, Ireland

JK, 0000-0002-3332-7064

Lameness in sheep is the biggest cause of concern regarding poor health and welfare among sheep-producing countries. Best practice for lameness relies on rapid treatment, yet there are no objective measures of lameness detection. Accelerometers and gyroscopes have been widely used in human activity studies and their use is becoming increasingly common in livestock. In this study, we used 23 datasets (10 non-lame and 13 lame sheep) from an accelerometer- and gyroscope-based ear sensor with a sampling frequency of 16 Hz to develop and compare algorithms that can differentiate lameness within three different activities (walking, standing and lying). We show for the first time that features extracted from accelerometer and gyroscope signals can differentiate between lame and non-lame sheep while standing, walking and lying. The random forest algorithm performed best for classifying lameness with an accuracy of 84.91% within lying, 81.15% within standing and 76.83% within walking and overall correctly classified over 80% sheep within activities. Both accelerometer- and gyroscope-based features ranked among the top 10 features for classification. Our results suggest that novel behavioural differences between lame and non-lame sheep across all three activities could be used to develop an automated system for lameness detection.

# 1. Introduction

Sheep production plays an important role in the future of food security as it provides both food (meat and milk) and fibre (wool) around the world [1]. Due to rising population level and consumer preferences, meat demand is expected to increase from 334 million tonnes in 2015 to 498 million tonnes in 2050 (https://www2.deloitte.com/content/dam/Deloitte/de/Documents/operations/Smart-livestock-farming_Deloitte.pdf). This means that farms need to increase productivity while maintaining highest standards of health and welfare. Health, welfare and production performance on farms can be improved by early detection and prompt treatment of diseased animals. Change in animal behaviour is often an early indicator of disease [2].

Lameness in sheep is the biggest health and welfare issue in most sheep-producing countries. In the UK, it costs between £24 and £80 million per annum [3] and in New Zealand costs excess of NZ$ 11 million (=£5.41 million approx.) annually [4] (significantly impacting on sheep, causing pain, discomfort, weight loss and therefore a reduction in production). Lameness in sheep is mainly a result of footrot (80%) [5], an infectious disease caused by an anaerobic bacteria *Dichelobacter nodosus*. Being an infectious disease, best practice regarding lameness in sheep relies on rapid treatment [6] based on early inspection and detection, followed by treatment with an injection of oxytetracycline or a similar antibiotic [6].

Detection of lame sheep is typically made based on visually identifying a change in locomotion by farmers that include particular features when walking such as nodding of the head and excessive flicking of head among others [5]. However, there are key challenges with this, as firstly, with the current global trend of large livestock farms [7] and large animal to staff ratio, visual observation of subtle and even more substantial behavioural signs might be impractical and sometimes impossible to detect; secondly, sheep are prey animals, they tend to mask signs of lameness in the presence of humans; and thirdly, walking (when typical visual lameness identification is done) only constitutes 2% [8] of the total activity of the day, so that makes the detection of lameness difficult. There are no studies so far that have explored if there are any behavioural features across activities that can differentiate lame and non-lame sheep.

Accelerometers and gyroscopes have been used widely in various livestock species, especially cattle, to automatically detect lameness. Despite this, no commercial solutions are available and the automatic classification of lameness in cattle is largely based on differences in the amount of activities in lame versus non-lame [9] animals, rather than on novel features in accelerometer and gyroscope signals that discriminate lameness. In comparison to cattle, there is limited work in sheep [10–12], with almost all studies focused on only classifying basic activities, such as standing, grazing and lying, without identifying different features that classify lameness. A recent study by Barwick *et al.* [12] using five sheep investigated the classification of lame walking activity in sheep; however, as mentioned above, walking only constitutes a very limited part of daily activity budget of sheep which limits its overall utility and does not give insight into the whole behaviour of a lame sheep.

In this study, we extend our previous algorithmic approach [11] to classify sheep activity, using an ear-based accelerometer and gyroscope sensor to investigate for the first time (i) if we could detect lameness in sheep using both accelerometer and gyroscope signals, (ii) which machine learning algorithms perform best at classifying lameness and what are the most important features for lameness classification and (iii) if we can classify lameness in sheep across a range of daily activities (walking, standing and lying).

# 2. Material and methods

For this study, a two-phase approach was developed, aiming at classifying sheep lameness. The first phase classified sheep activity, distinguished between walking, standing and lying. Once a sheep's activity was identified, a second classifier was applied in order to classify if a sheep is lame or not. Figure 1 illustrates this process. The first phase classification was performed using our previously published algorithms in Walton *et al.* [11], where we focused on the development of classification algorithms for standing, walking and lying in sheep, while the second phase classification was developed in the current study where we explicitly focus on discriminating between lame and non-lame samples within each of the different behaviours.

## 2.1. Study site and animals

Before starting the main trial, a pilot study for 2 days was conducted to check the research protocols described below and to give sheep some time to habituate to the sensor. Ethical permission was

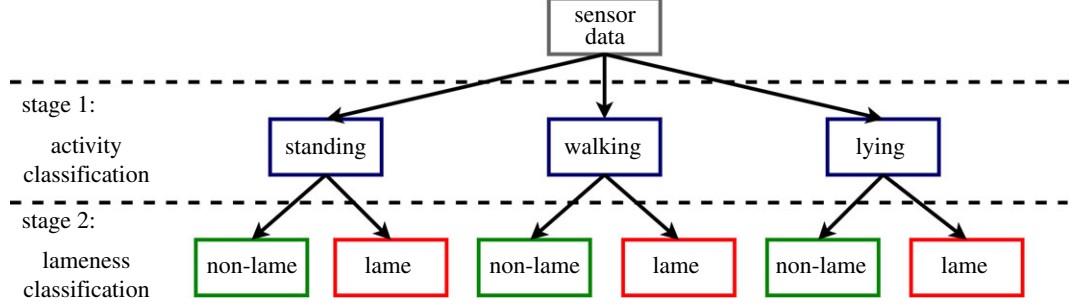

**Figure 1.** Two-phase classification approach.

obtained from the School of Veterinary Medicine and Science, University of Nottingham under unique number 1481 150603. During the trial, the Code of Research Conduct and Research Ethics from the University of Nottingham was followed. For the main trial in this study, data were collected in two periods: the first period was for 8 days in October 2016 and the second one was 4 days in October 2017. A total of 18 sheep (mixed breeds) for the whole study was selected for data collection, with 12 sheep selected in 2016 and 6 sheep selected in 2017. The selection was performed via stratified random sampling (age) from a flock of 140 animals at the University of Nottingham.

The breeds of sheep were Texel cross ($n = 10$), Suffolk cross ($n = 3$) and Mule ($n = 5$). The selected sheep had body condition scores ranging from 2.5 to 4, lameness scores ranging from 0 to 3 and age ranging from 1 to 4 years. Assessment of body condition score, age and breed was done at day 1 of each collection period. Body condition score (the degree of fatness of the living animal) was obtained using UK industry guidelines [13,14]. Sheep were kept on a pasture, a rectangular 0.3 acre field with 179.3 m perimeter in 2016 and in a rectangular 0.74 acre field with a 270 m perimeter in 2017. In both the fields, sheep had free access to water. In both periods of collection, sheep were sprayed with coloured livestock spray on either side of the sheep's body with a number to facilitate individual identification.

## 2.2. Data collection

Sensor data were collected using a custom-made wearable device based on the Intel® Quark™ SE microcontroller C1000. The device was designed to be used for data collection and classification and encompassed flash memory, a low-power-wide-area radio module and a Bosch BMI160 (Bosch-sensortec.com, 2016) low-power inertial measurement unit (IMU), featuring a 16-bit triaxial gyroscope and a 16-bit triaxial accelerometer. The optimal sampling rate, established in our previous work [11], was set at 16 Hz. The printed circuit board of the device had dimensions of $31.6 \times 35 \times 9$ mm (with parts mounted) and weighed 4 g. Sensors were attached to a 270 mA h Li-ion battery. The devices were attached to the sheep by fixing to the existing electronic identification ear tag using tape and lightweight plastic tie/Velcro. Devices were mounted on sheep at the beginning of each trial day at approximately 9.00 and removed the following morning at 9.00. Each day, sensors were prepared by annotating the switching time, and a time reference. The detailed procedure is described in [11]. Time synchronization for the solution was ultimately achieved using the radio to set the clock to the same as the host receiver.

## 2.3. Behavioural observations

Behaviours of interest for this study were three activities (walking, standing and lying) which were defined based on an ethogram in Walton *et al.* [11]. In this ethogram, walking was described as sheep moving forward in a four-beat motion for 2 s or more with head up and oriented in the direction of the movement, standing was described as when sheep was standing on their four legs with or without jaw movement, and lying as when sheep was lying on the ground in sternal or lateral recumbency with or without jaw movement. Lameness was scored visually by a trained observer based on a validated 0–6 locomotion score scale as described in Kaler *et al.* [15]. Ground truth behaviour observations were obtained from video recordings in 2016. Video footage of sheep behaviours was recorded using a handheld Panasonic HC-V380 video camera set to record time-stamped videos in an MP4 50 M format with 1080p ($1920 \times 1080$ pixels) quality. Time-stamped video recordings of the sheep were processed using Noldus Observer XT 11(Noldus) (www.noldus.com)

software to label the different behavioural categories. A total of 31.49 h of footage was observed in 2016. In 2017, ground truth observations were obtained from annotations made by NB (direct visual observation) using a digital stopwatch which was synchronized with devices using radio-based time synchronization. A total of 7.59 h of data were observed in 2017.

Lameness scores were grouped into two different classes: non-lame aggregating scores 0 and 1; and a lame group aggregating scores 2 and above based on best practice of classifying lameness 2 or above as lame [6]. These newly form groups (lame and non-lame) were used to obtain a binary classification of lameness.

## 2.4. Datasets

During the study, a total of 23 datasets (10 non-lame and 13 lame) with a sampling frequency of 16 Hz were collected from 18 different sheep. In total, there were 20 104 samples, each 7 s in duration. Of these samples, 31.57% (46.88% non-lame and 53.12% lame) of the samples included walking, 36.87% (51.56% non-lame and 48.44% lame) included standing and 31.56% (35.78% non-lame and 64.72% lame) included lying activity.

## 2.5. Data processing and features selection

Processing of the data was performed using custom-made scripts written in Python 3.5 [16]. First, the raw sensor data (accelerometer and gyroscope) and the ground truth behaviour information from both the video transcripts (2016 data) and manual recordings (2017 data) were aligned using the time stamps. Afterwards, each file was discretized into windows of 7 s with a 50% overlap between two consecutive windows [17]. Windows that had erroneous data (e.g. acceleration values of 0 for the whole window) were removed from the dataset. Lame status of the sheep was added as ground truth to the dataset.

For each time window, a set of feature characteristics [18] was extracted from the magnitude of the acceleration and the magnitude of the gyroscope which are defined by $\bar{A} = \sqrt{A_x^2 + A_y^2 + A_z^2}$ and $\bar{G} = \sqrt{G_x^2 + G_y^2 + G_z^2}$, where $A_x$, $A_y$, $A_z$, $G_x$, $G_y$, $G_z$ represent the acceleration and gyroscope signals at the axes $x$, $y$, $z$, respectively. The dynamic component of the acceleration and gyroscope sensors was removed by calculating the rate of change for both acceleration magnitude and gyroscope magnitude [19].

Table 1 shows the description of the 16 feature characteristics computed using both the acceleration magnitude difference and the gyroscope magnitude difference providing a total of 32 feature characteristics. In this study, we applied a feature selection process to order and rank features according to their importance. Feature selection in this study was carried out using a filter-based approach using ReliefF [20].

## 2.6. Lameness classification approach

Classification algorithms for lameness used in this study included random forest (RF), neural network (NN), support vector machine (SVM), AdaBoost and k-nearest neighbour (KNN). All algorithms were implemented using the 'scikit-learn' package [21] using the set of 32 previously described feature characteristics as input variables and the corresponding ground truth lameness and behaviour for each of the window samples as labels. The description of algorithms is provided in the electronic supplementary material. In the RF algorithm, the number of trees was set to 250, with a minimum sample count set for leaf nodes equal to 1, and with the number of features to consider when looking for the best split equal to the square root of the $n$ features. Kernel selection for the SVM was a radial basis function. For the NN algorithm, there was one layer with 100 nodes with a rectified linear unit function $f(x) = \max(0,x)$ as the activation function, and a learning rate of 0.0001. For the AdaBoost algorithm, the base learner was a decision tree with depth 1, the number of base learners was set to 100 with a learning rate of 1. For the KNN algorithm, $K$ was set to 5, the distance metric was Euclidean and no normalization was adopted during the training as it did not significantly improve overall performance.

An individual classifier model was developed within each of the different activities (walking, standing and lying). Classification performance was evaluated using 10-fold cross-validation, a commonly used methodology that provides a robust evaluation in classification model performance [22]. This technique split the original dataset into 10 subsets of equal size. Stratification was applied

**Table 1.** Feature characteristics computed using the change in the magnitude of the accelerometer and the magnitude of the gyroscope for each individual window. Here, $f$ represents the signal.

| features | description/formula |
| --- | --- |
| **time-domain features** | |
| standard statistical features | mean, standard deviation, minimum, maximum, skewness, kurtosis and interquantile range |
| zero crossings | number of zero crossings in a window after subtracting the window mean value |
| signal area | $SA = \sum Mag \cdot \dfrac{1}{f_s}$ <br> where $Mag$ is either accelerometer or gyroscope magnitude and $f_s$ is the sampling frequency |
| **frequency-domain features** | |
| spectral entropy | $SE = -\sum PSD_{norm}(f) \cdot \log(PSD_{norm}(f))$ <br> where PSD is the power spectral density computed from the discrete Fourier transform |
| dominant frequency | frequency at which the Fourier-transformed signal has its highest power |
| spectral area | $SpA = 2\sum_{n=1}^{N} S(f_n) \cdot \Delta f$ <br> $S(f_n)$: power spectral density at frequency $n$ |
| harmonic frequency (2nd and 3rd) | frequencies at which the signal has its second and third highest power values |
| harmonic ratio | ratio of the sum of the even amplitudes ($n = 2, 4, 6,\ldots$) to the sum of the amplitudes of the odd harmonics ($n = 1, 3, 5,\ldots$) from the discrete Fourier transform <br> $HR = \dfrac{\sum_{i=1} f_{2i}}{\sum_{i=1} f_{2i+1}}$ |

**Table 2.** Overview of the number of samples per behaviour, number of samples and ratios for non-lame and lame sheep in each of the behaviour classes.

| WSL | lameness | number of samples | sample ratio within behaviour (in %) |
| --- | --- | --- | --- |
| walking | non-lame | 2974 | 46.88 |
| | lame | 3370 | 53.12 |
| standing | non-lame | 3822 | 51.56 |
| | lame | 3591 | 48.44 |
| lying | non-lame | 2271 | 35.78 |
| | lame | 4076 | 64.22 |

during the splitting to ensure that class representations in each of the subsets were equal to the original dataset [23]. Stratification was based on the lameness class only. Then, over a total of 10 iterations (folds) are performed, and at each iteration, nine of the subsets were used to train a classification model, while the remaining one was held back as a test set. In each fold, the performance of the model built on the training set was evaluated using the test sets. After the 10 folds, every one of the 10 subsets had been used once as a test set, resulting in 10 sets of performance values, one for each fold. The average of these performance values represented the cross-validated classification performance. During the training of the data, no extra procedures were applied to balance the dataset as this was relatively well balanced for both activities and ratios for non-lame and lame samples in the three different activities, as shown in table 2.

The proportion of the data used for training and testing within each of the different models (walking, standing and lying) and for each individual sheep is reported in electronic supplementary material, tables S2–S4. In these tables, the average and standard deviation of the percentage of data are used

for each individual sheep. Scripts for the classification algorithm are provided in the electronic supplementary material.

## 2.7. Performance of the classification

The performance of the best performing algorithms out of all was evaluated using the metrics overall accuracy, precision, recall (also known as sensitivity), F-score and specificity as follows:

$$\text{overall accuracy} = \frac{TP + TN}{TP + TN + FP + FN},$$

$$\text{precision} = \frac{TP}{TP + FP},$$

$$\text{recall} = \frac{TP}{TP + FN},$$

$$F\text{-score} = 2 \times \frac{\text{precision} \times \text{recall}}{\text{precision} + \text{recall}}$$

and

$$\text{specificity} = \frac{TN}{TN + FP},$$

where TP (true positives) represents the number of instances where lameness was correctly classified by the algorithm and visually observed. FN (false negatives) represents the number of instances where lameness was visually observed, but was incorrectly classified as non-lame by the algorithm. FP (false positives) is the number of instances predicted as lame by the algorithm but observed as non-lame. TN (true negative) is the number of instances where the algorithm correctly classified a sample as non-lame when it was actually observed as non-lame. The F-score gives a measure of a test's accuracy and is the harmonic mean of precision and recall.

## 2.8. Individual sheep level $\sigma$-differences

In addition to the performance metrics described above, we computed an individual sheep level $\sigma$-difference using the ratio of samples classified as lame per individual sheep in the dataset. The $\sigma$-difference can be computed as follows:

$$\Delta_\sigma = (\mu_{\text{lame}} - \sigma_{\text{lame}}) - (\mu_{\text{non-lame}} + \sigma_{\text{non-lame}}),$$

where $(\mu_{\text{lame}}, \sigma_{\text{lame}})$ represents the mean and standard deviation of the per-sheep ratio of samples predicted as lame in the lame group and $(\mu_{\text{non-lame}}, \sigma_{\text{non-lame}})$ represents the mean and standard deviation of the per-sheep ratios of individual windows predicted as lame in the non-lame group. A significant difference ($p$-value less than or equal to 0.05) for the ratios of lameness between lame and non-lame was investigated using a Mann–Whitney $U$ test. When considering results at sheep level, an individual threshold for each of the activities was computed from the distribution of the percentage of lame samples. The threshold was computed by $(\mu_{\text{lame}} + \mu_{\text{non-lame}})/2$ which provided the maximum level of separation between the two groups.

# 3. Results

## 3.1. Lameness classification comparing algorithms

Figures 2 and 3 illustrate an example time series of the accelerometer and gyroscope magnitude differences (respectively) output for observed periods of lying, standing and walking among lame and non-lame sheep.

A comparison of classification accuracy for all the different algorithms and within the three different behaviours is shown in figure 4 for an unconstrained compute scenario.

When classifying lameness in walking, RF obtained the highest overall accuracies, followed by AdaBoost followed by KNN, when considering all activities. Besides RF and AdaBoost, other algorithms performed worse in terms of overall accuracy and their performance deteriorated as more features were added to the classification. The best accuracy, in an unconstrained scenario, was obtained using RF and 17 features, yielding an overall accuracy of 76.83%. In terms of overall

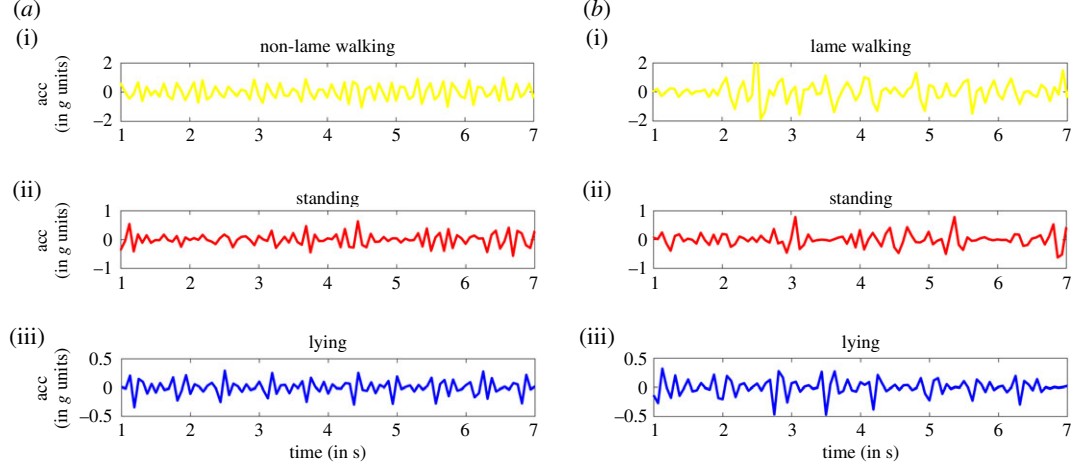

**Figure 2.** Accelerometer magnitude difference (in *g* units) for non-lame (*a*) and lame (*b*) for the three different behaviours: walking (i), standing (ii) and lying (iii). The plot was generated using a 7 s window.

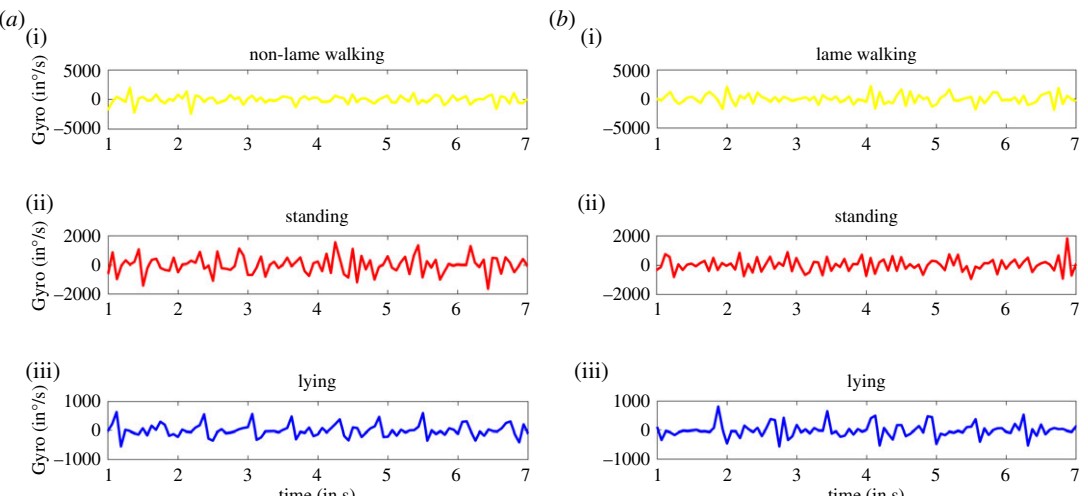

**Figure 3.** Gyroscope magnitude difference (in °/s units) for (*a*) non-lame and (*b*) lame for the three different behaviours: (i) walking, (ii) standing and (iii) lying. The plot is obtained for a 7 s window.

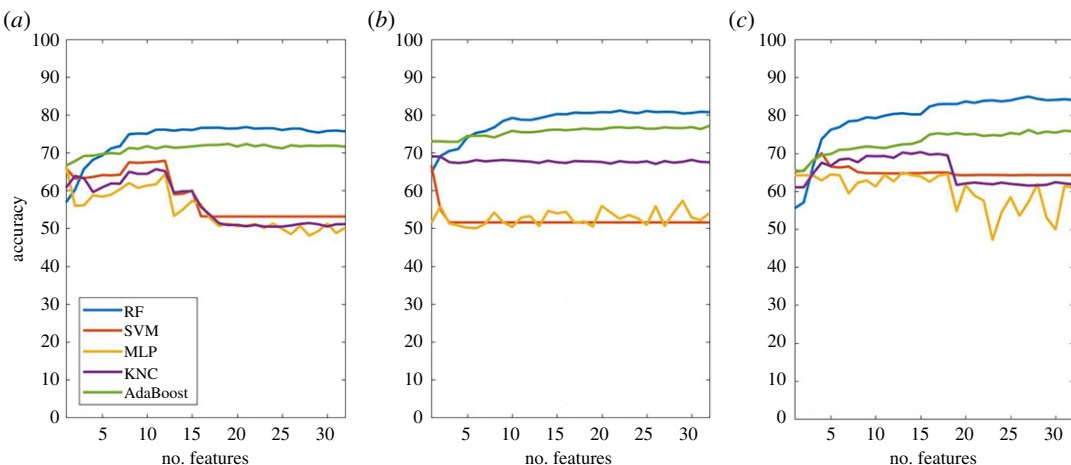

**Figure 4.** Comparison of the overall accuracy performance metric for lameness classification in (*a*) walking, (*b*) standing and (*c*) lying using the different learning algorithms.

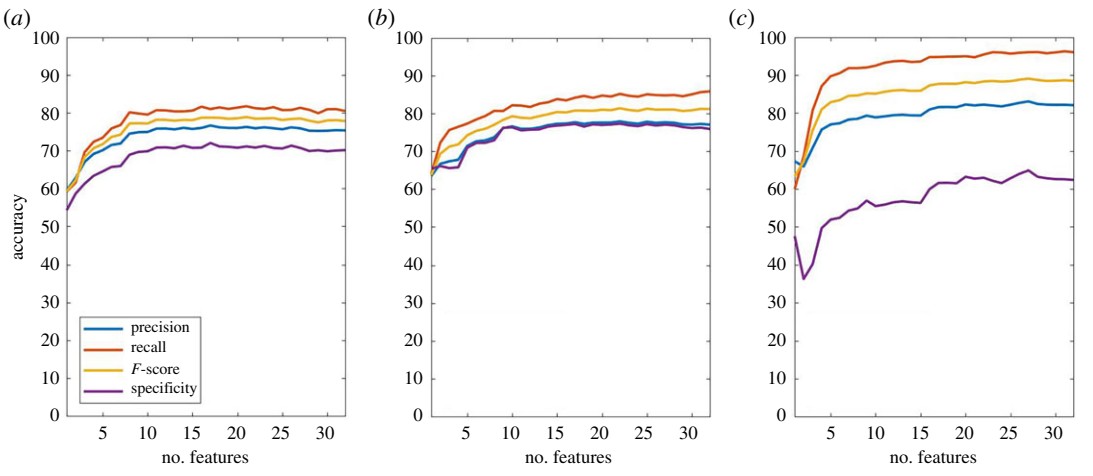

**Figure 5.** Performance metrics (precision, recall, *F*-score and specificity) for lameness classification in (*a*) walking, (*b*) standing and (*c*) lying.

accuracy, the RF algorithm started to plateau when more than eight feature characteristics were used, yielding overall accuracies between 74 and 76%.

Overall accuracies in standing were generally higher than the ones in walking. Once again, RF consistently outperformed the other algorithm types with accuracy peaking at 81.15% when using 22 features. Accuracy started to plateau between 80 and 81% once 15 or more features were used.

Classification of lameness within lying provided the highest accuracies compared to the other two behaviours, with RF yielding the best accuracy at 27 features with a value of 84.91%. Apart from the global maximum at 27 features, RF accuracy plateaued after 17 features between 83 and 84%. The difference between RF and AdaBoost performances overall was larger than those seen for walking and standing.

## 3.2. RF algorithm performance across activities

Detailed assessment of the performance of the classification, again in an unconstrained scenario, using the RF algorithm is shown in figure 5. For walking, all the different performance metrics of the classification increased with an increased number of features between 1 and 8 at which point all plateaued. After an initial increase, performance metrics did not show any significant changes. For standing, similarly all the different performance metrics of the classification increased with an increased number of features between 1 and 9 at which point all plateaued. For lying, precision recall and *F*-score metrics of the classification increased with an increased numbers of features between 1 and 5 at which point all of them plateaued with a small increase. Maximum performance values were obtained when using features 22, 32, 22 and 22 for precision, recall, *F*-score and specificity, respectively.

Of all the performance metrics, specificity was the lowest and recall was the highest across all activities.

## 3.3. Feature selection

Feature rankings were obtained using the ReliefF feature selection algorithm within each of the different behaviours (walking, standing and lying). The results of the top 10 feature ranking are shown in table 3. A table with the full 32 ranked features is provided in the electronic supplementary material, table S1.

For walking, 8 out the first 10 features were accelerometer based and 5 out of the first 10 are frequency-domain features. For standing, 6 out the first 10 were accelerometer-based features and 8 out of the first 10 were frequency-domain feature characteristics. For lying, out of the first 10 feature characteristics, 6 were accelerometer-based features and 5 were frequency-domain features.

## 3.4. Sheep level differences

The individual sheep level σ-differences obtained for walking, standing and lying using RF classifier with an increasing number of feature characteristics are shown in figure 6. Overall within lying, the

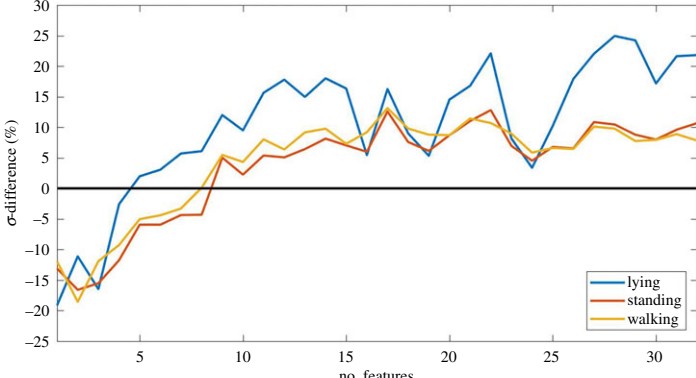

**Figure 6.** $\sigma$-differences for lying, standing and walking which represent the level of separation between the number of times a sheep was predicted as lame vs the number of times that it was predicted as non-lame. The black line indicates a $\sigma$-difference of 0.

**Table 3.** Top 10 ranked features using the ReliefF algorithm for walking, standing and lying. Light blue and dark blue colours represent acceleration magnitude difference-based features with frequency domain and time domain, respectively. Light and dark green colours represent gyroscope magnitude difference-based features with frequency domain and time domain, respectively.

| rank | walking | standing | lying |
|------|---------|----------|-------|
| 1 | spectral entropy | signal area | dominant frequency |
| 2 | zero crossings | interquartile range | spectral entropy |
| 3 | dominant frequency | spectral area | dominant frequency |
| 4 | dominant frequency | standard deviation | maximum |
| 5 | spectral entropy | zero crossings | standard deviation |
| 6 | 2nd harmonic frequency | dominant frequency | minimum |
| 7 | interquartile range | 2nd harmonic frequency | spectral entropy |
| 8 | signal area | spectral entropy | signal area |
| 9 | standard deviation | 3rd harmonic frequency | zero crossings |
| 10 | minimum | interquartile range | 2nd harmonic frequency |

large $\sigma$-differences were obtained with a 24.94% when using 28 features. A significant positive $\sigma$-difference was obtained using more than five features in lying, whereas in standing, a positive $\sigma$-difference was obtained for more than nine features, and for walking was obtained when using more than eight features.

Electronic supplementary material, tables S2–S4 show that the percentage of data used for each individual sheep ranged between 9 and 11% within walking and standing, and 10 and 13% within lying. As different numbers of samples were retrieved for each individual sheep, training and testing datasets contained more samples for some sheep than others. The cross-validation procedure applied ensured that no individual 7 s samples were used in both training and testing datasets, but since the data split was only stratified with regard to the lameness class, training and testing datasets in each cross-validation fold contained samples from the same sheep, causing data leakage at the sheep level. However, due to the random nature of how cross-validation folds are created, data leakage at the sheep level should not be biased towards a set of specific animals. The percentage of misclassified samples for each individual sheep within each of the activities (walking, standing and lying) is reported in the electronic supplementary material, tables S5–S7. The individual threshold for walking, standing and lying was when more than 54.93% were classified as lame for walking, more than 50% of the samples were classified as lame for standing and more than 73.45% were classified as lame for lying. Based on this, individual sheep classification led to

(i) 16 out of 18 were correctly classified providing a sheep level accuracy of 88.89% within walking.
(ii) 12 out of 15 were correctly classified providing a sheep level accuracy of 80.00% within standing.
(iii) 11 out of 12 were correctly classified providing a sheep level accuracy of 91.67% within lying.

# 4. Discussion

To the author's knowledge, this is the first study to demonstrate automated detection of lameness in sheep within three relevant activities (standing, lying and walking). For all of these activities, the study identified features that discriminated between lame and non-lame sheep. The novelty being that the study of signals within unobvious behaviours (i.e. non-walking activities) can identify lameness. This goes beyond the current knowledge that suggests that lameness can influence the activity budget of animals [9] (i.e. how much time they spend lying, etc.); as the results of this study suggest that lame sheep carry out activities differently leading to change in acceleration and rotational motion. In contrast to only detecting lameness within walking, as in [12], the ability to also detect lameness within lying and standing translates directly into a significantly higher sensitivity of detection since walking constitutes a small percentage of the total activity of the day [8]. In addition, the algorithm accuracy for classification of lameness within lying (85%) and standing (81%) was higher than those obtained with walking (76%). This could be possibly due to the level of noise in the signal while walking, which could be caused by dynamic movement (e.g. changing ground surface conditions in the field).

In the current study, detecting features that significantly differentiate lame from non-lame while walking was not a surprising result because of visual differences previously reported between the gait pattern of lame and non-lame sheep [24]. Within walking, five out of the six top feature characteristics were frequency-domain features which are linked to the rhythm and pace at which the walking movement is performed. Differences in frequency-domain features within walking might be the result of locomotion being limited by the disease in lame sheep compared to non-lame sheep, resulting in differences in the regularity and frequency of head movements. Spectral entropy, for example, represents the level of complexity of the behaviour and hence it translates into differences in the regularity of the movement. Lame sheep show alteration in gait with peculiar head nodding in line with stride compared to non-lame sheep which have a smoother stride pattern. In addition, differences in the dominant frequency can be translated to more or less head peak movements and rotations, which is again consistent with the head nod movements [15].

More interestingly, results for classification of lameness have higher accuracy within lying and standing activities. The top features included a mixture of frequency (e.g. dominant frequency, spectral entropy) and time-domain features (e.g. standard deviation, interquartile range). This suggests differences in the variability and smoothness of movements for both standing and lying between lame and non-lame sheep. For lame sheep, this could be an attempt to reduce discomfort as lameness can lead to significant pain [13], whereby lame sheep redirect their body weight to the unaffected leg leading to postural changes while standing; this has been noted in cows before [13], although no algorithms, to our knowledge, have been identified. For lying, existing research in cattle suggests that lame cows lie down for more time compared to non-lame cows; whereas our results suggest that lame sheep possibly lie differently than non-lame ones, and once again this could be due to animal's attempt to alleviate pain as mentioned above.

Considering implementation, relying solely on activity length (e.g. time lying above or below a threshold) as an indicator of lameness has inherent problems. For example, there could be a lot of confounding factors that affect the amount of activity, e.g. environmental conditions have shown to impact the amount of lying activity in non-lame cows [14]. The solution proposed by our study addresses the latter issue by enabling lameness detection via the classification of signals within an activity, not using activity length as a proxy. This approach also improves sensitivity given that a broader range of activities can be classified. Additionally, this offers more flexibility in terms of trade-offs between energy consumption versus classification accuracy as it is event based and readily lends to thresholding, i.e. one has degrees of freedom regarding sleep/energy consumption strategies for the overall device.

On a commercial device, other practicalities need to be considered, including the one on memory and energy consumption due to a solution based on activity length which log activities continuously. The central processing unit, which is an energy-expensive component, in such a scenario would be highly active thus offering less opportunity for setting the device on a sleeping mode. That said, the use of IMU thresholding could help mitigate the impact specifically if tied solely to one activity, e.g. walking.

Aside from acceptable classification accuracy, any proposed commercial implementation would need to consider the choice of algorithm, as well as the number and type of features used, because these factors significantly impact computational complexity and hence battery life leading to a shorter deployable life cycle of an offering. RF algorithms had the best accuracy in the off-line analysis and are relatively

computationally inexpensive when compared to NN, SVM and Bayes point machine. In the current study, using the ReliefF feature selection method, we determined that using the 10 highest ranked features is sufficient to detect lameness within the different activities without a significant impact on accuracy. Having less features could be advantageous for battery optimization as it can reduce computational complexity. However, it should be noted that at this point the study did not apply compute constraints and the type of feature is as important, if not more important, than the number of features given a constrained battery-powered deployment scenario. For example, frequency-domain features are more computationally expensive than time-domain features [25], and in the presented analysis, frequency-domain features ranked highly among the top 10 discriminative features. In short, with respect to a commercially deployable solution, further research is required to understand the performance of features and algorithms applied to a computationally and energy-constrained context.

Another key result from this study was that both accelerometer and gyroscope features were ranked among the top features. Most of the work for activity recognition in livestock has only used acceleration-based features, our study suggests that gyroscope features add value in improving the accuracy of the algorithms. This has also been reported in human activity recognition where gyroscope-based features have increased the accuracy of the algorithms for certain complex behaviours [26].

When comparing the different performance metrics of lameness detection across the different activities, although overall accuracy was relatively high with good precision, algorithms had relatively low specificity values (64.86–77.34%) suggesting a degree of false positives. This could be possibly due to confusion of classification of lameness with other possible behaviours such as grazing which can produce relatively high acceleration values. However, lameness classification in the current study was based on a 7 s sample window level, and results show that when comparing the ratio of lame and non-lame ratios at individual sheep level, the algorithm significantly differentiated lame and non-lame sheep. By aggregating the algorithms' results over multiple windows over the course of the day, one can enhance the sensitivity or specificity based on parallel or serial combination. In addition, this might be better for a deployable solution as it will reduce energy consumption related to data transmission [27].

One limitation of the current algorithm as presented in this study is that it only classifies between lame and non-lame sheep as a binary classification. Extension of the current lameness classification can be obtained to include all the different levels of lameness (scores 1–5) by incorporating more samples for each sheep in the correspondent levels of lameness severity. However, the classification between certain scores (e.g. 0 and 1) could be difficult to obtain with high accuracy due to subtle gait differences between them. Current 'best practice' in the UK recommends catching lame sheep at score 2 or above (the lameness classification was set to this in the current study). Thus, there might not be a need to differentiate lameness scores above 2 from lameness control purposes; however, algorithms able to differentiate more severe scores could be useful welfare indicators on the farm. Another possible limitation of the current algorithm is the existing data leakage at the individual sheep level, as in our algorithm design data were stratified only based on the lameness score and not sheep ID. However, the selection of the samples was not weighted towards a specific set of animals, and hence data leakage at the individual level should not affect the generalization of the algorithm. Moreover, at the individual sheep level, the algorithm shows very good levels with 88.89, 80 and 91.67% accuracy for walking, standing and lying, respectively.

Although our results fit in with epidemiology of lameness and provide novel insights, there is a need to validate these algorithms over an extended time period with larger group sizes, different breeds and fields/terrains to demonstrate the consistency of the detection within the three different activities both at the general and individual level [28]. Moreover, by monitoring over longer periods of time, it should be possible to investigate the evolution in the behavioural changes within the different activities as the lameness progresses in severity or when sheep start to show signs of recovery. This can then also be used to provide an assessment tool for the recovery or improved welfare under the different treatment practices. Given high animal welfare is on the agenda for all sheep-producing countries and there are no objective tools to measure lameness, we believe our study provides an important step towards this and our results could be incorporated to develop an automated and early-warning lameness detection system.

## 5. Conclusion

The results from this study show for the first time that there are novel behavioural differences between lame and non-lame sheep across three key behaviours: walking, standing and lying. These behavioural differences when captured via features generated from accelerometer and gyroscope signals could

differentiate between lame and non-lame sheep across all these three activities. The RF algorithm yielded the highest accuracy in the classification of lameness. These results could help in further development of an automatic system for lameness detection and help improve sheep health and welfare on farms.

Ethics. The study was reviewed and approved by the School of Veterinary Medicine and Science Ethics and Welfare Committee under the unique number 1481 150603.

Data accessibility. Data are deposited at the Dryad Digital Repository https://doi.org/10.5061/dryad.mk4fc3r [29].

Authors' contributions. J.K. conceived and designed the study. N.B., J.K. and J.A.V.-D. collected the data. K.A.E. did programming of the hardware and provided the materials for data collection. J.M., T.D. and J.K. did data curation and data visualization. J.M., J.K. and J.A.V.-D. analysed the data and drafted the manuscript. All authors reviewed and edited the manuscript and gave approval of the publication.

Competing interests. The authors declare that they do not have competing interests.

Funding. This work was supported by the Biotechnology and Biological Sciences Research Council (grant no. BB/N014235/1) and by Innovate UK (grant no. 132164).

Acknowledgements. We would like to thank Emma Gurney for her help with the field trial, Christy Casey (HPE) for his help with data pre-processing, and David Coates (Intel) for assistance on the Intel-based device developed for the project.

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
