## [Reviewer comments · Royal Society Open Science]

Review History

RSOS-190824.R0 (Original submission)

Review form: Reviewer 1

Is the manuscript scientifically sound in its present form?

Yes

Are the interpretations and conclusions justified by the results?

No

Is the language acceptable?

Yes

Do you have any ethical concerns with this paper?

No

Have you any concerns about statistical analyses in this paper?

Yes

Recommendation?

Accept with minor revision (please list in comments)

Comments to the Author(s)

See attached pdf for my comments (Appendix A).

Review form: Reviewer 2

Is the manuscript scientifically sound in its present form?

Yes

Are the interpretations and conclusions justified by the results?

Yes

Is the language acceptable?

No

Do you have any ethical concerns with this paper?

No

Have you any concerns about statistical analyses in this paper?

No

Recommendation?

Major revision is needed (please make suggestions in comments)

Comments to the Author(s)

The manuscript was enjoyable and certainly an interesting and important concept. The results are important for animal welfare scientists, the sheep-producing community and for those interested in precision livestock technology. I would like to see this published and hope that the comments in the attached file (Appendix B) will help to strengthen the paper.

Decision letter (RSOS-190824.R0)

27-Aug-2019

Dear Dr Kaler,

The editors assigned to your paper ("Automated detection of lameness in sheep using machine learning approaches: novel insights into behavioural differences among lame and non-lame sheep") have now received comments from reviewers. We would like you to revise your paper in accordance with the referee and Associate Editor suggestions which can be found below (not including confidential reports to the Editor). Please note this decision does not guarantee eventual acceptance.

Please submit a copy of your revised paper before 19-Sep-2019. Please note that the revision deadline will expire at 00.00am on this date. If we do not hear from you within this time then it will be assumed that the paper has been withdrawn. In exceptional circumstances, extensions may be possible if agreed with the Editorial Office in advance. We do not allow multiple rounds of revision so we urge you to make every effort to fully address all of the comments at this stage. If deemed necessary by the Editors, your manuscript will be sent back to one or more of the original reviewers for assessment. If the original reviewers are not available, we may invite new reviewers.

- Data accessibility

<http://datadryad.org/submit?journalID=RSOS&manu=RSOS-190824>

- Competing interests

- Authors' contributions

All submissions, other than those with a single author, must include an Authors' Contributions section which individually lists the specific contribution of each author. The list of Authors

should meet all of the following criteria; 1) substantial contributions to conception and design, or acquisition of data, or analysis and interpretation of data; 2) drafting the article or revising it critically for important intellectual content; and 3) final approval of the version to be published.

- Acknowledgements

- Funding statement

on behalf of Dr Manoj Srinivasan (Associate Editor) and Kevin Padian (Subject Editor)
openscience@royalsociety.org

Associate Editor's comments (Dr Manoj Srinivasan):

Both reviewers see substantial merit in the paper and would like to see it published, but request additional details so that the methods and results are presented in a more complete manner. We look forward to a revised article in due course.

Reviewers' Comments to Author:

Reviewer: 1

Comments to the Author(s)

See attached pdf for my comments.

Reviewer: 2

Comments to the Author(s)

The manuscript was enjoyable and certainly an interesting and important concept. The results are important for animal welfare scientists, the sheep-producing community and for those interested

in precision livestock technology. I would like to see this published and hope that the comments in the attached file will help to strengthen the paper.

Author's Response to Decision Letter for (RSOS-190824.R0)

See Appendix C.

RSOS-190824.R1 (Revision)

Review form: Reviewer 1

Is the manuscript scientifically sound in its present form?

Yes

Are the interpretations and conclusions justified by the results?

Yes

Is the language acceptable?

Yes

Do you have any ethical concerns with this paper?

No

Have you any concerns about statistical analyses in this paper?

No

Recommendation?

Accept as is

Comments to the Author(s)

I am very pleased with these revisions clarifying the generalizability of the model. I think the paper can be accepted in its current form.

Review form: Reviewer 2

Is the manuscript scientifically sound in its present form?

Yes

Are the interpretations and conclusions justified by the results?

Yes

Is the language acceptable?

Yes

Do you have any ethical concerns with this paper?

No

Have you any concerns about statistical analyses in this paper?

No

Recommendation?

Accept as is

Comments to the Author(s)

The authors have done a great job in addressing the comments and updating the manuscript. The paper will be of interest to many in academia and industry and I look forward to seeing it published.

Decision letter (RSOS-190824.R1)

29-Nov-2019

Dear Dr Kaler,

It is a pleasure to accept your manuscript entitled "Automated detection of lameness in sheep using machine learning approaches: novel insights into behavioural differences among lame and non-lame sheep" in its current form for publication in Royal Society Open Science. The comments of the reviewer(s) who reviewed your manuscript are included at the foot of this letter.

Kind regards,

Lianne Parkhouse

Editorial Coordinator

on behalf of Dr Manoj Srinivasan (Associate Editor) and Kevin Padian (Subject Editor)
openscience@royalsociety.org

Reviewer comments to Author:

Reviewer: 1

Comments to the Author(s)

I am very pleased with these revisions clarifying the generalizability of the model. I think the paper can be accepted in its current form.

Reviewer: 2

Comments to the Author(s)

The authors have done a great job in addressing the comments and updating the manuscript. The paper will be of interest to many in academia and industry and I look forward to seeing it published.

Appendix A

A summary of my review of “Automated detection of lameness in sheep using machine learning approaches: novel insights into behavioural differences among lame and non-lame sheep”:

In this paper, the authors use machine learning to detect lameness in sheep from body-worn sensor data. Compared to previous literature which has focused on walking, here they find that useful information needed for lameness classification is present when standing and lying in addition to walking. The paper is clearly written, choice of the model used is justified and the methodology is clearly outlined. It is good to see that the authors have conducted cross-validation. However, my main suggested revision is to (i) directly quantify how balanced the dataset is and (ii) to clearly quantify how much leakage there is between the training and test set. This is essential for drawing the right conclusions from the modeling results and for evaluating the generalizability of the model developed here — if researchers want to use them on a new dataset. Once my suggestion (detailed below) is addressed and if the code and data are made publicly available, the paper can be accepted in my opinion.

My suggestions for revision are:

- 1) Clear details about about how balanced the train and test data are is missing and is necessary to draw the right conclusions from the modeling results: i) % of train and test data from each sheep, iii) % of train and test sheep-wise data that are from a given activity i.e. walking vs lying vs standing, iv) % of misclassifications by the model that are for a given sheep and v) how many sheep were included in both train and test data (leakage). Without this information it is difficult to interpret the success of the model. For example, if all the misclassifications are for the data of 4 sheep then the actual success of the model is 9 out of 13 sheep. Or if a vast majority of the datapoints correspond to standing then the higher accuracy for standing classification is less surprising. Adding these simple details about the data would help the reader gauge how generalizable this model will be to their own dataset. The authors must also add a paragraph in the discussion section on how they think their model will generalize to new data.
- 2) The caption in figure 6 has typos: “lying, standing and lying” and “that was predicted as not been lame”. Please read the manuscript carefully once more for typos.
- 3) I am unable to access the data on dryad with the link provided.

Appendix B

Automated detection of lameness in sheep using machine learning approaches: novel insights into behavioural differences among lame and non-lame sheep

Many thanks for the opportunity to review this manuscript, which is timely in terms of both animal welfare and application of precision livestock technology. This paper presents a method to automatically detect the lameness in sheep using the data collected from an accelerometer and gyroscope ear sensor. The machine learning techniques are employed to classify the sheep into the lameness and non-lameness within three activities (standing, walking and lying), where the accelerometer and gyroscope based Features, time and frequency features are used in two-phase classifications. Overall, the paper is well structured, but lacks descriptions of some details of the method, especially that the key Hyperparameters in each classifier are not clarified. The experimental results are presented well. General comments and more detailed points of the main concerns for this manuscript are as follows:

General comments:

- The structure and grammar of sentences needs proofreading and correcting throughout. There are some places where the tense changes in the sentence and some sentences with extra words where they are not needed e.g. “with 12 in the 2016”.
- The introduction could benefit from more detail. The order of the introduction is also a little confusing. It feels like there are 5 main points to make: food security, the use of sheep, lameness, early detection, technology. The structure jumps a little between these ideas and makes it slightly difficult for the reader to follow.
- Please add your hypotheses and predictions to the end of the introduction.
- The Study site and animals section needs more detail. Please see the ARRIVE guidelines for some pointers of what could be added (<https://www.nc3rs.org.uk/sites/default/files/documents/Guidelines/NC3Rs%20ARRIVE%20Guidelines%202013.pdf>)
- In section 3, the authors mentioned that two-phase is composed of activities classification (first phase) and lameness classification (second phase), please highlight the novelty of the method presented in this manuscript compared to the previous work in [13].
- As one of data pre-processing steps, the dynamic component of the acceleration and gyroscope sensors was removed. It is desirable that the authors provide any other processing that was applied to de-noise the raw data and balance the training data?
- The authors have presented results to compare different classifiers according to the different number of features. The factor of features and their dimensionalities matters to the classification performance. However, it is not the only factor that could significantly affect the classifications. In the manuscript, some important details for each classifier need to be clarified. These details are critical which should be considered in the comparison of classifications. Has each classifier been optimized by hyperparameters tuning? It is desirable that the authors could provide more information related to classifiers. E.g. for RF, what are these parameters: the number of trees and the depth of each tree in the forest, the number of features to consider when looking for the best split. Kernel selection in the SVM. For the NN, what hyperparameters (number of layers, activation function, learning rate) are selected in the experiments. For the Adaboost, what are the base learner, the number of base learner and learning rate? For the KNN, what is the number of K, the distance metric and normalization adopted in the training?

Detailed comments:

Page 3

Line 13: As there are only 3 behaviours to describe, it would be useful to see the definitions within the paper.

Line 13-20: How much footage was observed?

Line 22-24: Will there be any plans to extend from non-binary to continuous in the future?

Please address any potential merits and feasibility in the discussion.

Line 28: suggestion: "...20104 samples, each 7 s in duration".

Line 44: The F-Score seems slightly less intuitive to decipher than the other metrics. Could you add a sentence to explain exactly what the F-score is measuring?

Page 7:

Lines 34-48: This is very wordy and a lot of the information is already readily seen in the figure. Consider condensing this paragraph.

The discussion is balanced and addresses the main points, the conclusions are valid based on the results.

Appendix C

University of
Nottingham
UK | CHINA | MALAYSIA

School of Veterinary Medicine and Science

University of Nottingham

Sutton Bonington

Leicestershire

LE12 5RD

Email Jasmeet.Kaler@nottingham.ac.uk

We thank the reviewers very much for their comments and believe that these comments have further improved the arguments and clarity of the manuscript.

Reviewer: 1

Reviewer(s)' Comments to Author:

Automated detection of lameness in sheep using machine learning approaches: novel insights into behavioural differences among lame and non-lame sheep

General comments:

1. The structure and grammar of sentences needs proofreading and correcting throughout. There are some places where the tense changes in the sentence and some sentences with extra words where they are not needed e.g. "with 12 in the 2016".

AU: Thanks for your comment. We have done minor editing suggestions in the main text.

2. The introduction could benefit from more detail. The order of the introduction is also a little confusing. It feels like there are 5 main points to make: food security, the use of sheep, lameness, early detection, technology. The structure jumps a little between these ideas and makes it slightly difficult for the reader to follow. Please add your hypotheses and predictions to the end of the introduction.

AU: Thanks for your comment. We have rearranged the structure and took some sentences out to make the introduction clearer.

3. The Study site and animals section needs more detail. Please see the ARRIVE guidelines for some pointers of what could be added

(<https://www.nc3rs.org.uk/sites/default/files/documents/Guidelines/NC3Rs%20ARRIVE%20Guidelines%202013.pdf>)

AU: Thanks for your comment. We have added more details to the study site and animals

4. In section 3, the authors mentioned that two-phase is composed of activities classification (first phase) and lameness classification (second phase), please highlight the novelty of the method presented in this manuscript compared to the previous work in [13].

AU: Thanks for your comment. There was a typo with the reference it should be [12]. We have corrected this, and we have highlighted the novelty of our method compared to previous work [12].

5. As one of data pre-processing steps, the dynamic component of the acceleration and gyroscope sensors was removed. It is desirable that the authors provide any other processing that was applied to de-noise the raw data and balance the training data?

AU: Thanks for your comment. Apart from the removing the dynamic component of the acceleration and gyroscope we only removed data files that showed erroneous information (i.e., dataset consistently showing an acceleration of 0 and similarly for data files with erroneous gyroscope). No method was applied for the balancing of the training dataset since the dataset was relatively well balance as shown Tables 2 with ratio of lame vs non-lame samples.

6. The authors have presented results to compare different classifiers according to the different number of features. The factor of features and their dimensionalities matters to the classification performance. However, it is not the only factor that could significantly affect the classifications. In the manuscript, some important details for each classifier need to be clarified. These details are critical which should be considered in the comparison of classifications. Has each classifier been optimized by hyperparameters tuning? It is desirable that the authors could provide more information related to classifiers. E.g. for RF, what are these parameters: the number of trees and the depth of each tree in the forest, the number of features to consider when looking for the best split. Kernel selection in the SVM. For the NN, what hyperparameters (number of layers, activation function, learning rate) are selected in the experiments. For the Adaboost, what are the base learner, the number of base learner and learning rate? For the KNN, what is the number of K, the distance metric and normalization adopted in the training?

AU: Thanks for your comment. Information on the classifiers has been added to the text. It is as follows: a) in RF the number of trees was 250, with a minimum sample count set for leaf nodes equal to 1, and with the number of features to consider when looking for the best split equals to the square root of the n features; b) the kernel selection for the SVM was a Radial Basis Function (RBF); c) for the NN there was one layer with 100 nodes with a rectified linear unit function $f(x) = \max(0, x)$ as activation function, and a learning rate of 0.0001; d) for the Adaboost the base learner was a decision tree with depth 1, the number of base learners was set to 100 with a learning rate of 1; e) for the KNN, K was set to 5, the distance metric was Euclidean and no normalization was adopted during the training as it did not significantly improve overall performance.

Detailed comments:

Page 3

Line 13: As there are only 3 behaviours to describe, it would be useful to see the definitions within the paper.

AU: Thanks for your comment. Added

Line 13-20: How much footage was observed?

AU: Thanks for your comment. This has been added into the main text. A total of 31.49 hours of footage was observed in 2016. A total of 7.59 hours of data were observed in 2017.

Line 22-24: Will there be any plans to extend from non-binary to continuous in the future? Please address any potential merits and feasibility in the discussion. Need to add this in the discussion.

AU: This has been added.

Line 28: suggestion: "...20104 samples, each 7 s in duration".

AU: Thanks for your comment. Corrected

Line 44: The F-Score seems slightly less intuitive to decipher than the other metrics. Could you add a sentence to explain exactly what the F-score is measuring?

AU: Thanks for your comment. Added

Page 7:

Lines 34-48: This is very wordy and a lot of the information is already readily seen in the figure. Consider condensing this paragraph. The discussion is balanced and addresses the main points, the conclusions are valid based on the results.

AU: Thanks for your comment. We have condensed this.

Reviewer: 2

A summary of my review of “Automated detection of lameness in sheep using machine learning approaches: novel insights into behavioural differences among lame and non-lame sheep”:

In this paper, the authors use machine learning to detect lameness in sheep from body-worn sensor data. Compared to previous literature which has focused on walking, here they find that useful information needed for lameness classification is present when standing and lying in addition to walking. The paper is clearly written, choice of the model used is justified and the methodology is clearly outlined. It is good to see that the authors have conducted cross validation. However, my main suggested revision is to (i) directly quantify how balanced the dataset is and (ii) to clearly quantify how much leakage there is between the training and test set. This is essential for drawing the right conclusions from the modeling results and for evaluating the generalizability of the model developed here — if researchers want to use them on a new dataset. Once my suggestion (detailed below) is addressed and if the code and data are made publicly available, the paper can be accepted in my opinion.

My suggestions for revision are:

1) Clear details about how balanced the train and test data are is missing and is necessary to draw the right conclusions from the modeling results: i) % of train and test data from each sheep, iii) % of train and test sheep-wise data that are from a given activity i.e. walking vs lying vs standing, iv) % of misclassifications by the model that are for a given sheep and v) how many sheep were included in both train and test data (leakage). Without this information it is difficult to interpret the success of the model. For example, if all the misclassifications are for the data of 4 sheep then the actual success of the model is 9 out of 13 sheep. Or if a vast majority of the datapoints correspond to standing then the higher accuracy for standing classification is less surprising. Adding these simple details about the data would help the reader gauge how generalizable this model will be to their own dataset. The authors must also add a paragraph in the discussion section on how they think their model will generalize to new data.

AU: Many Thanks for your helpful comments. Specific points are clarified below

- The balance of the dataset is in Tables 2 in the main text for the number of lame vs non-lame within each individual activity.
- The percentage of test data from each sheep is reported in Tables 2-4 in the Supplementary material. Each table correspond to different behaviour and we have added in the main text a line to clarify that for each of the activities a classification model was constructed in case that was unclear before
- The percentage of misclassified samples for each individual sheep within each activity it is now reported in Tables 5-7 in the Supplementary Material. And also added this to results

We have added a paragraph on the generalization of the model in the discussion. We have also added the code into the supplementary material as requested.

2) The caption in figure 6 has typos: “lying, standing and lying” and “that was predicted as not been lame”. Please read the manuscript carefully once more for typos.

AU: Thanks for your comment. Fixed for figure 6 and we have checked and corrected for other typos.

3) I am unable to access the data on dryad with the link provided.

AU: Thanks for your comment. We have double checked with dryad for the accessibility of the data it should now be working.